# Unveiling the sensory and interneuronal pathways of the neuroendocrine connectome in *Drosophila*

**Sebastian Hückesfeld[1], Philipp Schlegel[2], Anton Miroschnikow[1], Andreas Schoofs[1], Ingo Zinke[1], André N Haubrich[3], Casey M Schneider-Mizell[4], James W Truman[4], Richard D Fetter[4], Albert Cardona[4,5,6], Michael J Pankratz[1]\***

[1]Department of Molecular Brain Physiology and Behavior, LIMES Institute, University of Bonn, Bonn, Germany; [2]Department of Zoology, University of Cambridge, Cambridge, United Kingdom; [3]Life & Brain, Institute for Experimental Epileptology and Cognition Research, University of Bonn Medical Center Germany, Bonn, Germany; [4]Janelia Research Campus, Howard Hughes Medical Institute, Ashburn, United States; [5]MRC Laboratory of Molecular Biology, Cambridge Biomedical Campus, Francis Crick Avenue, Cambridge, United Kingdom; [6]Department of Physiology, Development and Neuroscience, Cambridge, United Kingdom

**Abstract** Neuroendocrine systems in animals maintain organismal homeostasis and regulate stress response. Although a great deal of work has been done on the neuropeptides and hormones that are released and act on target organs in the periphery, the synaptic inputs onto these neuro-endocrine outputs in the brain are less well understood. Here, we use the transmission electron microscopy reconstruction of a whole central nervous system in the *Drosophila* larva to elucidate the sensory pathways and the interneurons that provide synaptic input to the neurosecretory cells projecting to the endocrine organs. Predicted by network modeling, we also identify a new carbon dioxide-responsive network that acts on a specific set of neurosecretory cells and that includes those expressing corazonin (Crz) and diuretic hormone 44 (Dh44) neuropeptides. Our analysis reveals a neuronal network architecture for combinatorial action based on sensory and interneuronal pathways that converge onto distinct combinations of neuroendocrine outputs.

**\*For correspondence:**
pankratz@uni-bonn.de

**Competing interest:** The authors declare that no competing interests exist.

## Introduction

An organism is constantly subject to changing environmental challenges to homeostasis, and it counteracts these changes by adapting its physiology and behavior (*Selye, 1956*). In order to regulate homeostasis, animals must sense and integrate external and internal changes and generate outputs that comprise fundamental motivational drives such as feeding, fleeing, fighting, and mating (*Pribram, 1960*). This output ultimately leads to motor activities through movement of muscles or through secretion of hormones that act on target tissues. The neuroendocrine system in any animal with a nervous system plays a vital role in controlling both forms of outputs. In its simpler form, for example, in cnidarians, this takes place in a single sheet of epidermal cells that subsumes the functions of sensory, inter-, motor neurons and peptidergic cells (*Grimmelikhuijzen et al., 1996*; *Martin, 1992*). With more complex systems, the requirement for environmental sensing, integrating information and controlling motor outputs has given rise to specialized neurons of the periphery and the central nervous system (CNS) (*Buijs and Van Eden, 2000*; *Hartenstein, 2006*; *Toni, 2004*).

In mammals, different hormonal axes exist to keep essential physiological functions in balance, including the hypothalamic-pituitary-adrenal (HPA), the hypothalamic-pituitary-thyroid, the

somatotropic, and the two reproductive axes (*Charmandari et al., 2005*; *Fliers et al., 2014*; *Grattan, 2015*; *Kaprara and Huhtaniemi, 2018*). The various neuroendocrine axes also regulate each other. For example, the stress regulatory HPA axis relies on corticotropin releasing hormone (CRH) in the hypothalamus and has a negative influence on the reproductive regulatory axis (hypothalamic-pituitary-gonadal [HPG]) by inhibiting gonadotropin releasing hormone (GnRH) (*Kageyama, 2013*; *Rivier et al., 1986*) such that when nutrients are scarce, the reproductive system is negatively affected until metabolic homeostasis is re-established (*Tilbrook et al., 2002*). The peptidergic basis for homeostatic regulation has also been characterized in invertebrates. These include, to name a few, stress (*Johnson and White, 2009*; *Kubrak et al., 2016*; *Veenstra, 2009*), metabolism and growth (*Cannell et al., 2016*; *Dus et al., 2015*; *Gáliková et al., 2018*; *Geminard et al., 2006*; *Kahsai et al., 2010*; *Kim and Rulifson, 2004*; *McBrayer et al., 2007*), and development (*Hartenstein, 2006*; *Jindra et al., 2013*; *Truman, 2019*; *Truman et al., 1981*; *Wigglesworth, 1965*). For comprehensive reviews, see *Nässel, 2018*, *Nässel and Winther, 2010*, and *Nässel and Zandawala, 2019*. Despite the extensive characterization of the neuroendocrine system in both vertebrates and invertebrates, very little is known regarding the sensory inputs to the neuroendocrine cells in the CNS. In general, a neuroendocrine system consists of neurosecretory cells in the brain that release peptides/hormones into the circulation to modulate effector organs. Via hormonal feedback loops, the neuroendocrine system is able to tune its regulatory function to set itself back into homeostasis. However, the synaptic pathways of sensory signals onto the neurosecretory cells in the brain remain largely elusive.

The *Drosophila* larva is a well-suited model to tackle the issue of the sensory pathways that act on the central neuroendocrine system. Parallels to the mammalian HPA system have been pointed out at physiological and anatomical levels. The pars intercerebralis (PI) and pars lateralis (PL) regions of the larval brain are considered to be analogous to the vertebrate hypothalamus. The three known endocrine glands (collectively known as the ring gland) – the corpora cardiaca (CC), the corpus allatum (CA), and the prothoracic gland (PG) – exert functions that are physiologically similar to the vertebrate pituitary gland (*de Velasco et al., 2007*; *Hartenstein, 2006*; *Scharrer and Scharrer, 1944*). These produce the critical metabolic, growth, and maturation factors that are released directly into the circulation (adipokinetic hormone from the CC; juvenile hormone from the CA; ecdysone from the PG). There are also analogies in basic functional and anatomical features that interconnect the hypothalamus and the brainstem in mammals, and the PI/PL region and the subesophageal zone (SEZ) in

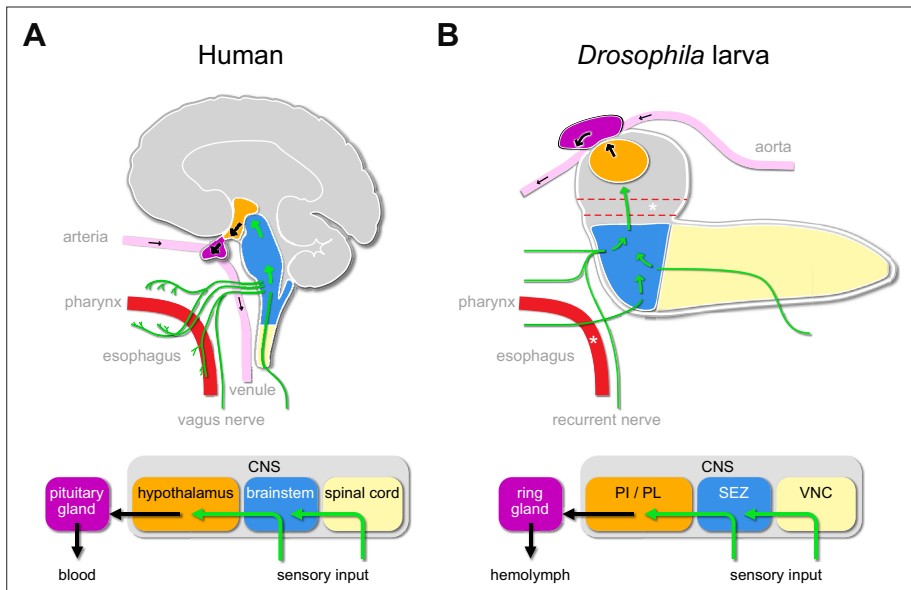

**Figure 1.** Sensory to endocrine pathways. (**A**) Schematic showing information flow from sensory input (green) to the endocrine system in the human brain compared to the *Drosophila* larval brain. (**B**) Asterisks denote the foramen (dotted red tube), where the esophagus (solid red tube) would pass through the CNS. Green arrows denote flow of sensory information; black arrows denote release of hormones into the circulatory system. CNS: central nervous system; PI: pars intercerebralis; PL: pars lateralis; SEZ: subesophageal zone; VNC: ventral nerve cord.

*Drosophila.* These also include the connections from the enteric nervous system to the CNS via the vagus nerve in mammals and the recurrent nerve in *Drosophila* (*Schlegel et al., 2016*; *Schoofs et al., 2014b*; *Figure 1*).

Leveraging a synaptic resolution serial section transmission microscopy (ssTEM) volume of a whole first instar larval CNS (*Eschbach and Zlatic, 2020*; *Miroschnikow et al., 2020*; *Ohyama et al., 2015*; *Schlegel et al., 2017*; *Thum and Gerber, 2019*; *Vogt, 2020*), together with functional analysis of the hugin neuropeptide circuit, we have been characterizing the neuronal circuits that control specific aspects of feeding behavior and the sensorimotor pathways of the pharyngeal nerves that drive food intake (*Hückesfeld et al., 2016*; *Miroschnikow et al., 2018*; *Schlegel et al., 2016*; *Schoofs et al., 2014a*). We now provide a comprehensive analysis of all neurosecretory cells that target the ring gland and the sensory neurons that form synaptic contacts with these cells, either directly or through interneurons. The neuronal network is organized in parallel interneuronal pathways that converge onto distinct combinations of neurosecretory cells based on different sensory inputs. The circuit architecture allows variable and flexible action to maintain homeostasis and growth in response to broad multi-sensory and diverse metabolic signals. Using network modeling, we also identify novel carbon dioxide ($CO_2$)-responsive sensory pathways onto a specific set of neuroendocrine outputs.

## Results

### EM reconstruction of the neuroendocrine system

To elucidate the sensory inputs onto the neuroendocrine cells, we first reconstructed the ring gland and the interconnected portion of the aorta (AO), and all neurons that project to these structures (*Figure 2A*). Reconstruction of a subset of the neurons in the PI was described earlier (*Schlegel et al., 2016*). All neurosecretory cell clusters found previously by light microscopy analysis (*Siegmund and Korge, 2001*) were identified and compared to expression patterns of known peptide-Gal4 driver lines. Since cell clusters that project to the ring gland (we collectively refer to them as ring gland projection neurons [RPNs]) have varying names, we use here the peptide names that these neurons are mainly known for (*Figure 2B*, *Figure 2—figure supplement 1*). CA-LP1 and CA-LP2 neurons were the only ones for which we could not unambiguously identify the neuropeptide identity, but found their expression in two independent Burs-Gal4 lines; also FMRFamide-positive projections were found in the CA, which likely are derived from the CA-LP1 or CA-LP2 neurons (*de Velasco et al., 2007*). To analyze ion transport peptide (ITP) neurons (*de Haro et al., 2010*; *Herrero et al., 2007*; *Kahsai et al., 2010*), we generated LexA-knock-in lines (*Figure 2—figure supplement 2*). A comprehensive overview for all RPN clusters analyzed in this study is provided in *Figure 2—source data 1*.

### Peptidergic and synaptic outputs

Peptidergic signaling is accomplished through release from dense core vesicles (DCVs). The specific peptidergic output region of all cells was identified by contacts of DCVs with the membrane with the apparent liberation of small dense particles, as exemplified for the corazonin neurons (CRZ) (*Figure 2C*). The outputs of all 10 peptidergic RPN groups are restricted mainly to the CC and AO. PTTH and CA-LP project almost exclusively to the PG and CA, respectively (*Figure 2D*). Neurons producing the stress-related peptide Crz (*Kubrak et al., 2016*) showed the broadest output pattern, targeting all tissues (*Figure 2C and D*). We also analyzed the reliability of determining the output release site by quantifying DCV fusions sites. Using Crz and corazonin receptor (CrzR)-expressing cells as an example, we could confirm by trans-Tango system (*Inagaki et al., 2012*) that the CC cells are the main target of CRZ (*Figure 2—figure supplement 3*). Thus, DCVs found in the PG or CA might mean that other RPNs, like PTTH and CA-LP, express the CrzR (for PTTH shown in *Imura et al., 2020*). These data further lend support that DCV fusion sites represent a reliable measure for targets of RPNs. The anatomical data on peptidergic outputs were combined with existing single-cell transcriptomic data on the larval brain (*Brunet Avalos et al., 2019*). Focusing on the expression of neuropeptides and their cognate receptors within the ring gland system, we confirmed, for example, that CRZ are targeting all other RPNs by releasing Crz as well as short neuropeptide F and proctolin (*Figure 2—figure supplement 4*). At the same time, the Crz receptor is expressed in the CC and to a lesser extent in the PG and CA, as well as in other RPNs. Based on the peptides and receptors expressed by the distinct RPN groups, the analysis uncovers complex interactions between neuroendocrine cells.

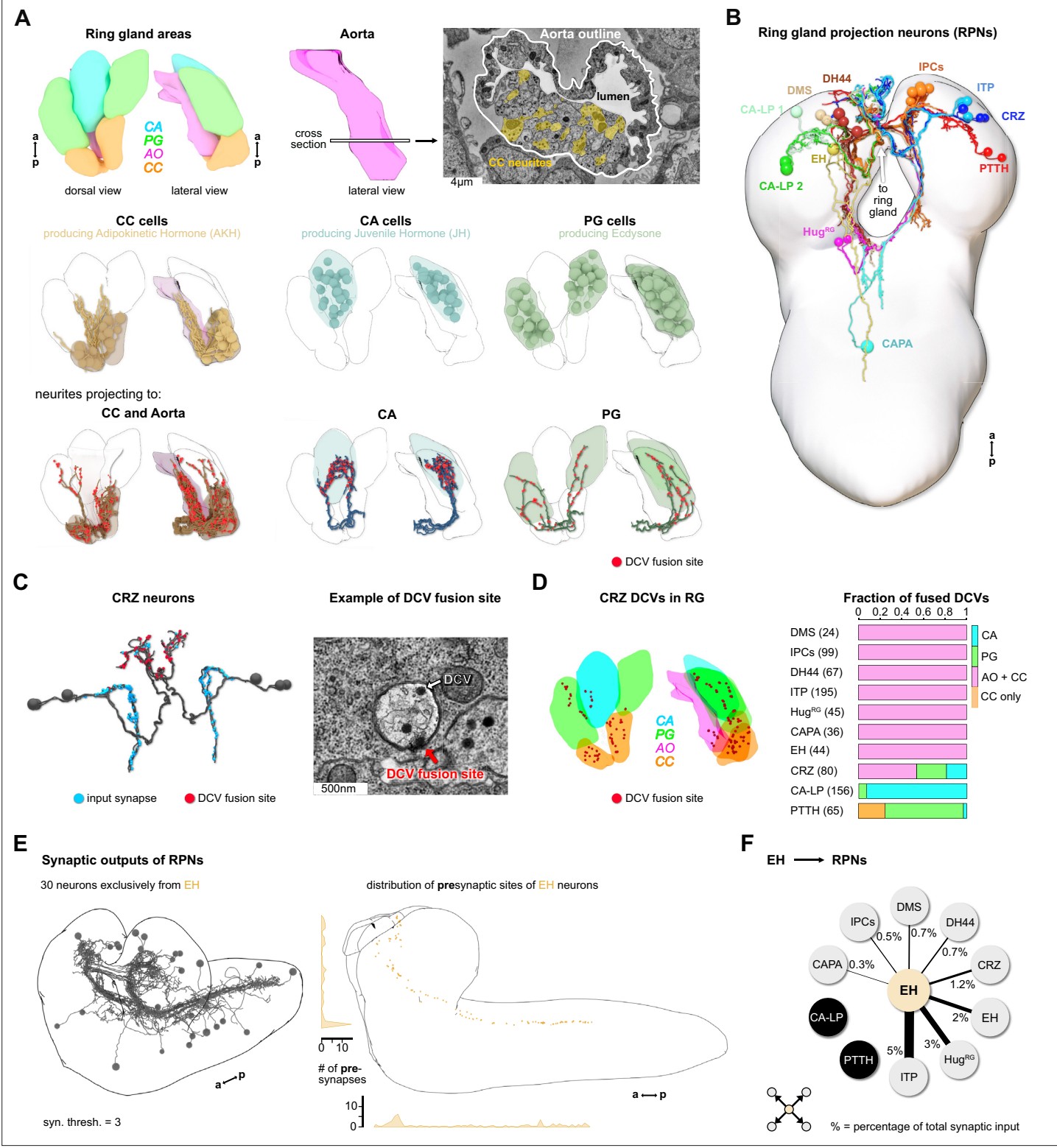

**Figure 2.** Reconstruction of the *Drosophila* larval ring gland and RPNs. (**A**) *Upper panel*: 3D reconstructed RG areas in dorsal and lateral view (CC = orange, PG = green, CA = blue, AO = pink). Cross section of the AO: colored areas represent single neurites of different CC cells. *Middle panel*: dorsal and lateral view of the RG showing the different cells in the distinct RG areas (CC, CA, and PG). *Lower panel*: neurites innervating the RG areas were separated based on innervation of the CC and aorta, only CA or only PG. Fused DCVs are marked as red dots. (**B**) Schematic of all 56 neurons innervating the RG named by the main neuropeptide produced. Total number of neurons per RPN cluster: DMS = 4, IPCs = 14, DH44 = 6, CRZ = 6, ITP

*Figure 2 continued on next page*

*Figure 2 continued*

= 8, CA-LP = 6, PTTH = 4, HugRG = 4, CAPA = 2, EH = 2. For clarity, only one side is shown for each neuronal cluster. (**C**) *Left*: reconstructed CRZ. Fused DCVs were marked as non-polar output synapses at distal neurites in RG tissues (red dots). Blue dots represent chemical synaptic input sites. *Right*: example picture of a DCV fusion site in the EM volume (DCV has to be fused to the membrane). (**D**) *Left*: magnification of the reconstructed RG with spatial distribution of CRZ DCV fusion sites (red dots).*Right*: quantification of all DCV fusion sites found in the RG areas for each RPN group. Numbers in brackets are total numbers of marked DCVs. The X-axis represents a fraction of fused DCVs. (**E**) *Left*: synaptic outputs of all RPNs (threshold = 3 synapses) constitute in total 30 neurons, which are exclusively downstream of EH RPNs. *Right*: spatial distribution of presynaptic sites of EH. EH neurons are the only RPNs having presynaptic sites located along abdominal, thoracic segments, and SEZ and protocerebrum. (**F**) EH neurons synaptically target other RPNs. Percentage represents the fraction of input of distinct RPNs from EH neurons, for example, ITP neurons receive 5% of its inputs from EH. a: anterior; AO: aorta; CA: corpus allatum; CA-LP: corpus allatum innervating neurosecretory neurons of the lateral protocerebrum; CAPA: capability neurons; CC: corpora cardiaca; CRZ: corazonin neurons; DCVs: dense core vesicles; DH44: diuretic hormone 44 neurons; DMS: *Drosophila* myosuppressin neurons; EH: eclosion hormone neurons; Hug$^{RG}$: Hugin neurons innervating ring gland; IPCs: insulin-producing cells; ITP: ion transport peptide neurons; p: posterior; PG: prothoracic gland; PTTH: prothoracicotropic hormone neurons; RG: ring gland; RPN: ring gland projection neurons; SEZ: subesophageal zone; ssTEM: serial section transmission electron microscope.

The online version of this article includes the following source data and figure supplement(s) for figure 2:

**Source data 1.** Comprehensive overview of all *Drosophila* RPN clusters.

**Figure supplement 1.** Neurons projecting to the ring gland in *Drosophila* larvae.

**Figure supplement 2.** CRISPR/Cas-dependent integration of T-LEM cassettes into *ITP* intron.

**Figure supplement 3.** Fused DCV sites as proxy for real release sites in the ring gland: example CRZ.

**Figure supplement 4.** Analysis of larval brain transcriptome data.

**Figure supplement 5.** Thresholds for reconstruction and analysis of the RPN connectome.

At this point, it is unclear to what extent these peptide-receptor interactions occur between peptides released within the CNS or found in the hemolymph.

We next addressed the issue of the largely unknown synaptic connectivity of the neuroendocrine cells by reconstructing the synaptic up- and downstream partners of all RPNs (threshold of three synapses to each RPN). For information on completeness of our analysis and the criteria for choosing certain threshold values, see *Figure 2—figure supplement 5*. We identified 30 downstream partners that, unexpectedly, were exclusively targeted by the two eclosion hormone neurons (EH), one on each side of the ventromedial protocerebrum (*Figure 2E*). The functional significance of the EH synaptic outputs is as yet unknown. However, it has been shown that the neurohemal release sites could be removed and the axon stumps electrically stimulated; this evoked an ecdysis motor program through interaction of the EH with response circuitry in the ventral nerve cord (VNC) (*Hewes and Truman, 1991*). Notably, these include all the other RPNs with the exception of CA-LP and PTTH, which regulate the activity of two major growth/maturation hormones, namely juvenile hormone and ecdysone (*Figure 2F*).

## Synaptic inputs onto the neuroendocrine system

We identified 209 upstream partners of the RPNs, whose synaptic sites are distributed in the anterior thoracic and SEZ region and up along the protocerebrum in a sprinkler-like fashion (*Figure 3A*). Unlike the RPNs in the PI (IPCs, DMS, DH44), which have significant amounts of monosynaptic connections with sensory neurons (*Miroschnikow et al., 2018*; *Schlegel et al., 2016*), the RPNs of the PL (CRZ, PTTH, CA-LP, and ITP) have no direct sensory input. Similarly, EH, CAPA, and Hug$^{RG}$ (hugin neurons innervating ring gland) RPNs have only small amounts of direct sensory contacts (*Figure 3—figure supplement 1*). We therefore focused on the interneurons and their connection with the sensory system.

We first divided the upstream interneurons into two groups: interneurons receiving direct sensory input and those that do not (threshold at two synapses); slightly more than half of all upstream neurons integrate sensory information, n = 110 (*Figure 3—figure supplement 1*). Based on previous publications, we know the peripheral origin (e.g., enteric, pharyngeal, olfactory) of most sensory neurons (*Berck et al., 2016*; *Miroschnikow et al., 2018*). Here, we additionally characterize a subset of tracheal dendritic neurons (TD neurons) (*Qian et al., 2018*; *Schlegel et al., 2016*) as being responsive to $CO_2$ levels (*Figure 3—figure supplement 2*). To determine which sensory signals are integrated by RPNs via these interneurons, we grouped their sensory inputs based on their peripheral origin (*Figure 3B*). The resulting map provides a comprehensive overview of the sensory to endocrine pathways in the

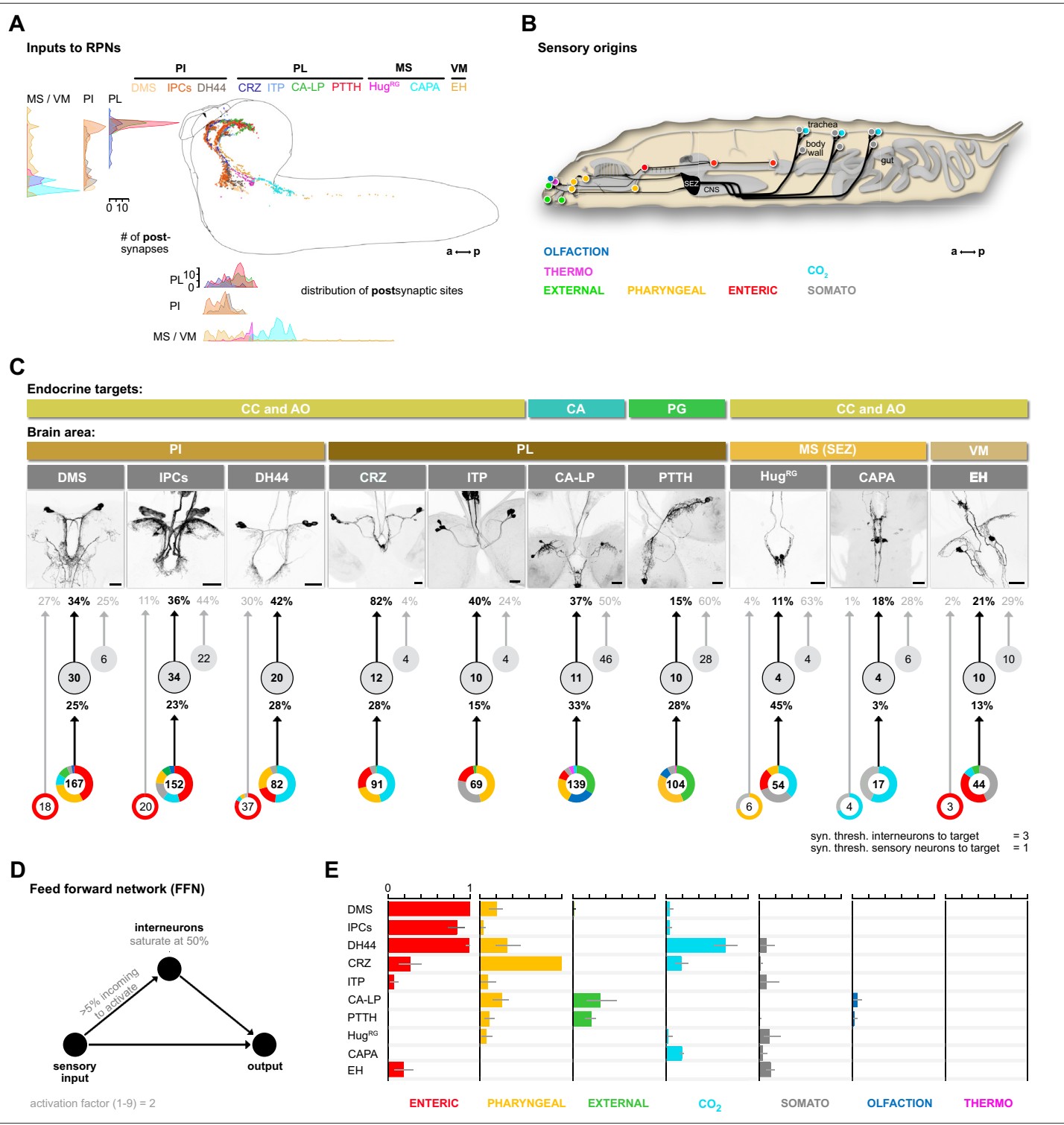

**Figure 3.** Inputs of RPNs and sensory origins. (**A**) Spatial distribution of postsynaptic sites of all RPNs (color coded). RPN postsynaptic sites are located along upper SEZ and in the protocerebrum in a sprinkler-like fashion. (**B**) Schematic side view of a *Drosophila* larva. Colored dots represent the location of sensory organs, based on their sensory origin. (**C**) Synaptic connections to RPNs (grouped) *from top to bottom*: RPNs are grouped by their endocrine targets or their location of somata within the CNS (brain area, colored bars). RPNs (displayed by expression pattern of the respective Gal4 or LexA lines) receive synaptic inputs (fraction of total synaptic inputs as percentage) from distinct sets of interneurons (numbers in circles represent the number of interneurons connected to RPNs), which in turn receive information from sensory neurons (fraction of total synaptic input as percentage). Colored pie charts represent the sensory profile through which interneurons (grouped) of each RPN group integrate sensory inputs (numbers in white

*Figure 3 continued on next page*

*Figure 3 continued*

circles). Colors of pie charts correspond to the respective sensory origins shown in (**B**). Note that the monosynaptic sensory neurons are also involved in polysynaptic pathways to the RPNs. (**D**) Scheme of the FFN. Sensory neurons are activated with an activation factor of 2 in the FFN. When more than 5% of presynaptic neurons are active, interneurons become activated up to an activity of 50%. (**E**) Summary of sensory-driven modulation of RPN output groups by FFN. The X-axis for each panel shows the mean activity of RPNs listed on the Y-axis. Colors represent the different sensory origins used to activate the network through 1- and 2-hop synaptic connections. a: anterior; AO: aorta; CA: corpus allatum; CA-LP: corpus allatum innervating neurosecretory neurons of the lateral protocerebrum; CAPA: capability neurons; CC: corpora cardiaca; CNS: central nervous system; $CO_2$: carbon dioxide; CRZ: corazonin neurons; DCVs: dense core vesicles; DH44: diuretic hormone 44 neurons; DMS: *Drosophila* myosuppressin neurons; EH: eclosion hormone neurons; FFN: feed forward network; Hug$^{RG}$: hugin neurons innervating ring gland; IPCs: insulin-producing cells; ITP: ion transport peptide neurons; MS (SEZ): medial subesophageal ganglion; p: posterior; PG: prothoracic gland; PI: pars intercerebralis; PL: pars lateralis; PTTH: prothoracicotropic hormone neurons; RPNs: ring gland projection neurons; syn. thresh.: synaptic threshold; VM: ventromedial cells.

The online version of this article includes the following figure supplement(s) for figure 3:

**Figure supplement 1.** Connectivity of the distinct sensory origins to RPN upstream neurons.

**Figure supplement 2.** A subset of TD sensory neurons responds to $CO_2$.

**Figure supplement 3.** Parameter changes in the FFN.

**Figure supplement 4.** Parameter changes in the FFN.

**Figure supplement 5.** Parameter changes in the FFN.

**Figure supplement 6.** Adjacency matrix of all neurons used in this study.

larval neuroendocrine system (*Figure 3C*). All of the RPNs receive input from a distinct combination of interneurons, which in turn receive input from a distinct combination of sensory neurons. In one extreme (e.g., IPCs), 152 sensory neurons from six different sensory regions (greatest from enteric) target 34 interneurons. At the other extreme (e.g., CAPA), 17 sensory neurons from two sensory regions target just four interneurons. The synaptic load of RPNs from interneurons that receive sensory inputs varies greatly. The largest is for CRZ, where 82% (fraction of input synapses) of the total input is from interneurons with direct sensory connections.

## Modeling the impact of activating sensory neurons on the neuroendocrine system

To assess the potential impact of sensory inputs on the neuroendocrine system, we employed a network diffusion model based on direct monosynaptic and 2-hop polysynaptic connections using feed-forward connectivity (*Figure 3D*). The model is deliberately kept simple as we lack detailed knowledge on the physiology (e.g., neurotransmitter) of the neurons involved. Such networks have been recently used successfully in the mouse to model sensory impact on activity in higher brain centers of the thalamus (*Shadi et al., 2020*). Our model predicts the impact of specific sensory origins onto each RPN group (*Figure 3E*; for parameterization and connection types in the model, see *Figure 3—figure supplements 3–5*; adjacency matrix for all neurons used in this study in *Figure 3—figure supplement 6*). As a first experimental analysis based on the predictions, we chose the $CO_2$ path because the defined sensory organ, that is, TD neurons, and distinct modality ($CO_2$) made it more tractable.

## A novel $CO_2$-dependent trachea to endocrine pathway

The model predicts a strong impact of TD ($CO_2$) neurons on DH44, CRZ, DMS, and CAPA RPNs (*Figure 3E*). To validate this, we performed imaging experiments using the ratiometric calcium integrator CaMPARI-2 to measure changes in activity of the RPNs upon $CO_2$ exposure (*Figure 4*). Indeed, the in vivo experiments confirmed the predictions for DH44 and CRZ RPNs, which were strongly activated by $CO_2$ (*Figure 4*). Weaker activation of DMS and IPCs was also observed, consistent with the predicted weak effects. CAPA neurons did not differ significantly from control groups but tended to show a lower activity upon $CO_2$ stimulation. Since the network diffusion model does not take the sign of a connection into account, it is conceivable that CAPA neurons are inhibited by $CO_2$. The analysis of connectivity based on the EM volume enabled us to identify a new circuit in which $CO_2$ level is detected by TD neurons, integrated by a core set of four thoracic interneurons (somata located in T1–T3 segments), which in turn strongly connect to DH44 and CRZ (*Figure 5A and B*). Each of the thoracic interneurons have slightly different connectivity profiles in terms of their up- and downstream partners (*Figure 5C*). Thus, while all four are interconnected to $CO_2$ sensory neurons and target DH44

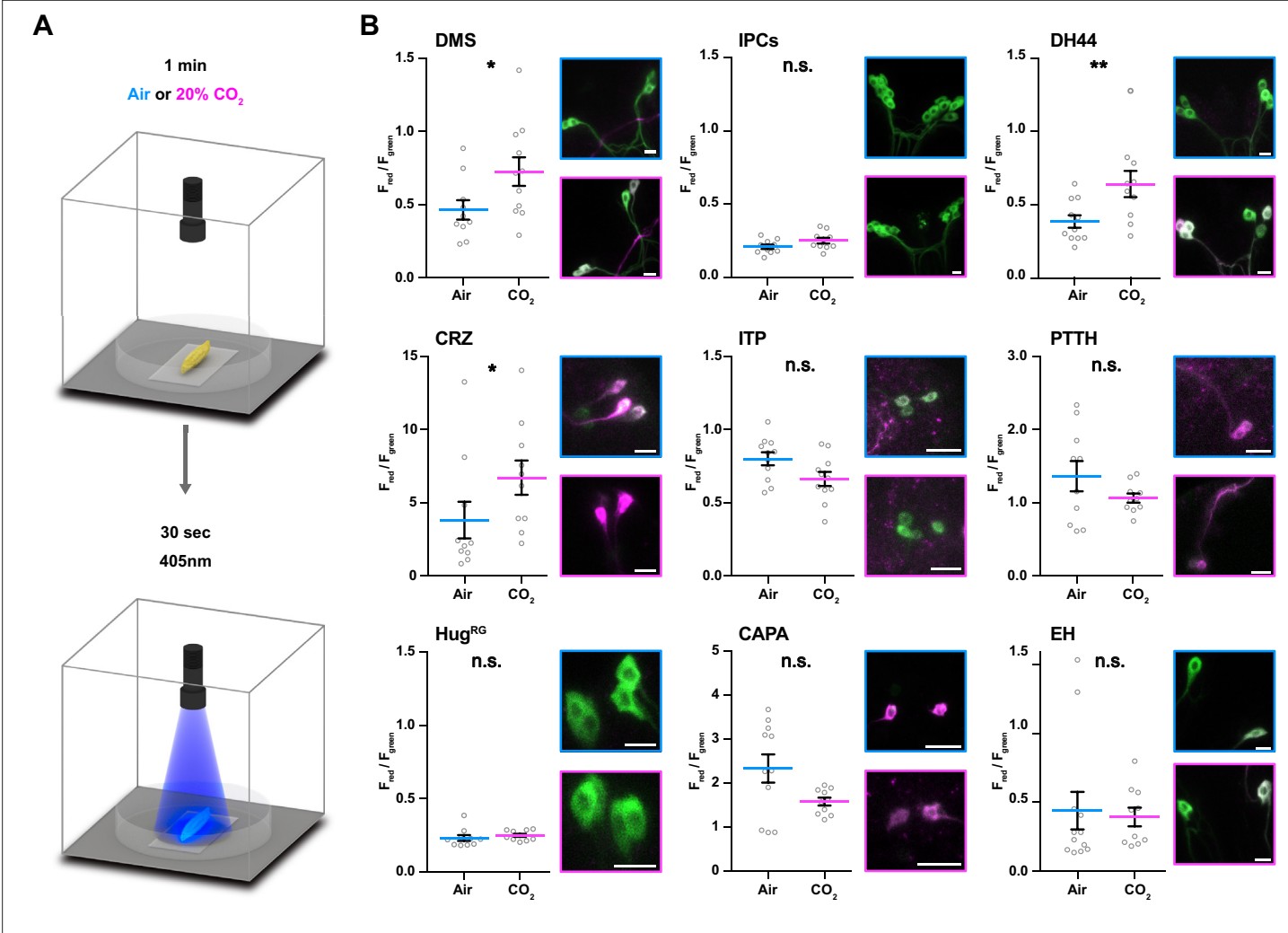

**Figure 4.** Impact of $CO_2$ stimulation on RPNs. (**A**) Setup for $CO_2$ stimulation of intact larvae. After 1 min of $CO_2$ exposure (0 or 20%), light of 405 nm wavelength was activated for 30 s. (**B**) Different peptide Gal4-lines driving expression of CaMPARI-2 in RPN clusters. Note that certain peptidergic clusters show baseline activity (CRZ, PTTH, CAPA) and therefore different scaling for the Y-axis was used, which represents the red to green fluorescence ratio. Significant activity changes could be observed for DMS, DH44, and CRZ upon $CO_2$ stimulation (magenta bars) compared to air (blue bars). Images next to the graphs show representative maximum projections of imaged cells (blue border = air, magenta border = 20% $CO_2$). All scale bars represent 20 µm. CaMPARI-2: calcium-modulated photoactivatable ratiometric integrator 2; CAPA: capability neurons; $CO_2$: carbon dioxide; CRZ: corazonin neurons; DH44: diuretic hormone 44 neurons; DMS: *Drosophila* myosuppressin neurons; EH: eclosion hormone neurons; Hug$^{RG}$: hugin neurons innervating ring gland; IPCs: insulin-producing cells; ITP: ion transport peptide neurons; n.s.: not significant; PTTH: prothoracicotropic hormone neurons; RPN: ring gland projection neuron.

or CRZ, the strength of the connections differs as well as their connections to other sensory neurons and RPNs. Please see *Figure 5—figure supplement 1* and 2 for identity (ID number and connectivity) of all interneurons.

We then took the two main output RPNs of the tracheal $CO_2$-responsive circuit (CRZ and DH44) and asked what other interneurons were upstream of these, and to which sensory neurons these interneurons were connected (*Figure 5D*). For CRZ, the strongest are in fact not the thoracic interneurons from the $CO_2$ pathway: one hemilateral pair of interneurons (#10, Munin 2) accounts for over 50% of total synaptic input to the CRZ neurons. These interneurons receive sensory information exclusively from pharyngeal sensory neurons (*Figure 5D*, top hive plot). There are two other strongly connected interneurons (#9, Munin 1; #12, subesophageal zone into brain neuron [SiB]), and they receive most of their inputs from the enteric region. Furthermore, all the interneurons are also part of pathways that target several RPNs. For example, interneuron #10 targets all neurons of the PL, whereas interneuron

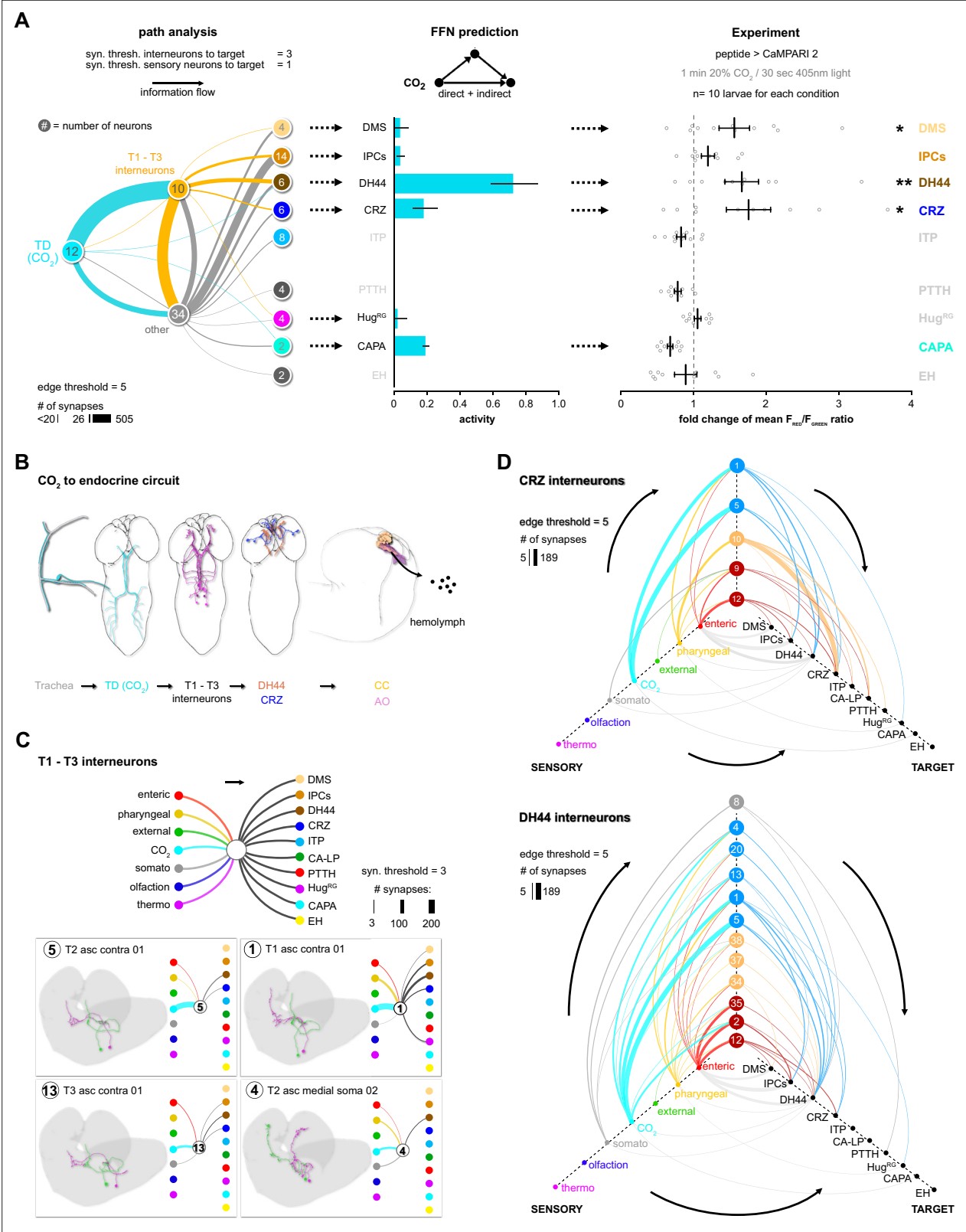

**Figure 5.** $CO_2$-dependent pathway from TD neurons to RPNs. (**A**) Comparison of underlying connectivity of TD ($CO_2$) neurons via interneurons to the RPNs, with the predicted outcome of mean activity (with an activation factor of 2; when more than 5% of presynaptic neurons are active, interneurons become activated up to an activity of 50%) of RPNs, and the outcome of CaMPARI-2 $CO_2$ experiments. FFN diffusion model reliably shows modulation of the RPNs. Please note that the circled numbers in the path analysis refer to the total number of neurons, not neuron identification number. (**B**)

*Figure 5 continued on next page*

*Figure 5 continued*

Using the combination of connectivity, prediction, and functional imaging experiments, a new sensory to endocrine neural circuit can be derived. TD ($CO_2$) neurons at the trachea respond to $CO_2$ levels and communicate predominantly via a core set of thoracic interneurons to DH44 and CRZ, which show release sites in the CC and AO. (**C**) Connectivity of the single thoracic interneurons (hemilateral pairs) to presynaptic sensory origins and to the distinct postsynaptic RPN groups. Thoracic interneurons receive additionally other sensory modalities apart from TD ($CO_2$) neurons and target different combinations of RPNs. (**D**) CRZ interneurons: hive plot showing the polysynaptic pathways from all sensory origins to all RPN target groups using the interneurons (synaptic threshold = 3) that target CRZ. Main sensory origins are enteric, pharyngeal, and $CO_2$. DH44 interneurons: TD ($CO_2$) represent the most dominant polysynaptic path from sensory origins to DH44. Note that monosynaptic connections from sensory neurons to RPNs are shown in gray. AO: aorta; CA-LP: corpus allatum innervating neurosecretory neurons of the lateral protocerebrum; CaMPARI-2: calcium-modulated photoactivatable ratiometric integrator 2; CAPA: capability neurons; CC: corpora cardiaca; $CO_2$: carbon dioxide; CRZ: corazonin neurons; DH44: diuretic hormone 44 neurons; DMS: *Drosophila* myosuppressin neurons; EH: eclosion hormone neurons; FFN: feed forward network; Hug$^{RG}$: hugin neurons innervating ring gland; IPCs: insulin-producing cells; ITP: ion transport peptide neurons; PTTH: prothoracicotropic hormone neurons; RPNs: ring gland projection neurons; ssTEM: serial section transmission electron microscope; TD: tracheal dendritic neurons.

The online version of this article includes the following figure supplement(s) for figure 5:

**Figure supplement 1.** Neuron catalogue of interneurons receiving sensory input.

**Figure supplement 2.** Neuron catalogue of interneurons receiving sensory input.

**Figure supplement 3.** Impact of Gr21a candidate neurons on modulation of RPN activity.

#12 targets all neurons of the PI. For DH44, the strongest upstream partners are the same thoracic interneurons that respond to $CO_2$ (**Figure 5D**, bottom hive plot).

Currently, one chemosensory receptor, gustatory receptor 21a (Gr21a), has been shown to be responsive to $CO_2$ (**Faucher et al., 2006**). It is expressed in the terminal organ, which is located in the anterior part of the larvae, but not in the TD neurons (**Figure 3—figure supplement 2**). Although we cannot determine the precise projections of the Gr21a-expressing neurons in the EM volume, a connectivity analysis of the chemosensory projections of the terminal and ventral organs (we cannot differentiate between projections originating from TO and VO in the EM volume) reveals only very weak connections to the DH44 or CRZ neurons, suggesting that $CO_2$-responsive Gr21a terminal organ and TD neuronal pathways are largely independent (**Figure 5—figure supplement 3**).

In sum, this illustrates the distinct sensory-to-neuroendocrine connectivity profiles (which sensory origins onto which set of RPNs) of the different interneurons.

## Interneurons that direct sensory information to distinct sets of neuroendocrine outputs

We next extended the connectivity hub analysis to the other interneurons of the neuroendocrine system (**Figure 6**). In the first approach, we plotted the sensory-to-interneuron-to-target paths for each RPN (**Figure 6A**). Shown are examples from RPNs located at different regions of the CNS, and one can see the large variation in the number and type of interneurons present that are directly connected to the sensory neurons. For instance, the IPCs receive inputs from the largest number of such interneurons, and from these, interneuron #11 (Hugin$^{PC}$, hugin neurons innervating protocerebrum) provides the largest input. At the other extreme, Hug$^{RG}$ neurons receive inputs from the least number of such interneurons; however, one of these, namely #17 (Dpilp7), provides 8% of total input that the Hug$^{RG}$ neurons receive. The analysis also illustrates the wide range of differences in synaptic strength between sensory neurons and interneurons as compared to between interneurons and target RPNs. For example, interneuron #1 has strong connections to sensory neurons but weak connections to the IPCs; by contrast, interneuron #16 has weak connections to sensory neurons but strong connections to the IPCs. A similar situation is observed between the interneurons #5 (one of the thoracic interneurons) and #10 (Munin 2) in terms of targeting CRZ.

In the second approach, we calculated the fraction of sensory inputs to given interneurons and multiplied it with the fraction of inputs of the RPN (**Figure 6B**). This analysis revealed interneurons that play a major role in the sensory pathways to the neuroendocrine system. Selected notable interneurons are illustrated in **Figure 6C**. For example, both #11 (Hug$^{PC}$) and #12 (SiB) interneurons have their strongest inputs from the enteric sensory neurons; however, whereas Hug$^{PC}$ interneurons strongly target just the IPCs (edge threshold of minimum five synapses), SiB interneurons target DMS, IPCs, and CRZ (**Figure 6C**).

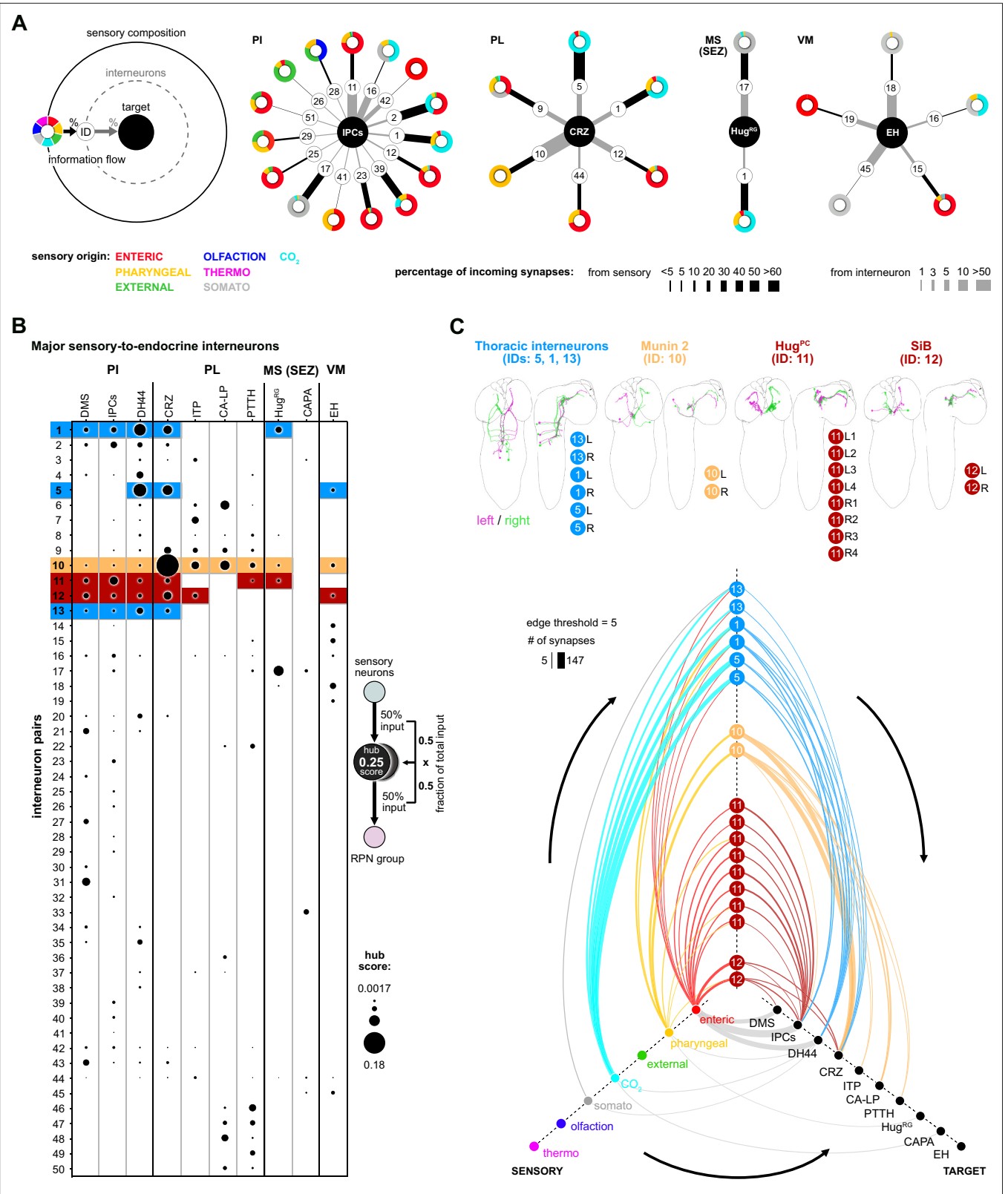

**Figure 6.** Interneurons and hub analysis of sensory to endocrine pathways. (**A**) Schematic of graph representation. Outer ring represents the sensory composition of neurons targeting upstream neurons of RPNs. Synaptic threshold for upstream neurons of RPNs = 3. Line thickness to interneurons and targets represents the percentage of synaptic input. Striped ring represents the interneuron layer (black lined white circle). Inner ring represents target neurons (RPN peptide clusters). Sensory-interneuron circuits of neurons within the PI region. IPCs integrate mainly information from enteric

*Figure 6 continued on next page*

*Figure 6 continued*

sensory areas. Sensory-interneuron circuits within the PL region. CRZ integrates mainly $CO_2$ and pharyngeal sensory information. Peptides of the MS (SEZ) or VM cluster show least number of sensory-interneuron input. Hug$^{RG}$ neurons receive sensory information from two interneuron pairs integrating mainly somatosensory and $CO_2$ sensory information. Please note different scaling for strength of connections between sensory origin to interneurons (black lines) and interneurons to target peptide groups (gray lines). (**B**) Dot plot showing the importance of interneurons acting as sensory to endocrine hub. Dot size was calculated using the fraction of total input an interneuron receives from sensory neurons multiplied by the fraction of total input this interneuron gives to an RPN output group. Colored backgrounds of dots are highlighted for (**C**). (**C**) Selected interneurons (highlighted in **B**) connecting the sensory system with RPNs. Thoracic interneurons receive sensory information from TD ($CO_2$) neurons and target IPCs, DH44 and CRZ (hive plot, strongest connection = 147 synapses). Munin 2 interneurons connect CRZ, ITP, PTTH, and CA-LP RPNs with pharyngeal sensory neurons. Hug$^{PC}$ connect the IPCs with enteric sensory neurons. SiB neurons also receive information from enteric origins but target DMS, IPCs, and CRZ. Edge threshold for hive plot = 5 synapses. CA-LP: corpus allatum innervating neurosecretory neurons of the lateral protocerebrum; CAPA: capability neurons; $CO_2$: carbon dioxide; CRZ: corazonin neurons; DH44: diuretic hormone 44 neurons; DMS: *Drosophila* myosuppressin neurons; EH: eclosion hormone neurons; Hug$^{PC}$: hugin neurons innervating protocerebrum; Hug$^{RG}$: hugin neurons innervating ring gland; IPCs: insulin-producing cells; ITP: ion transport peptide neurons; MS (SEZ): medial subesophageal ganglion; PI: pars intercerebralis; PL: pars lateralis; PTTH: prothoracicotropic hormone neurons; RPNs: ring gland projection neurons; SiB: subesophageal zone into brain neurons; VM: ventromedial.

The online version of this article includes the following figure supplement(s) for figure 6:

**Figure supplement 1.** Sensory-interneuron paths to specific RPNs.

**Figure supplement 2.** Sensory integration by CA-LP and PTTH RPNs.

**Figure supplement 3.** Olfactory pathways to neuroendocrine cells.

There are also intriguing unique groups, for example, the interneurons (#s 46–50), which are highly specialized for CA-LP and PTTH (*Figure 6B*, *Figure 6—figure supplement 2*); these receive strong sensory inputs from the olfactory system (for a comprehensive connectivity map, see *Figure 6—figure supplement 3*). In adult *Drosophila*, it was shown that the release of juvenile hormone from the CA potentiates sensitivity of a pheromone sensing olfactory receptor OR47b (*Lin et al., 2016*) to maximize courtship success of male flies. In larvae, we found several previously appetitive and aversive assigned olfactory receptor neurons (*Kreher et al., 2008*) being connected via multiglomerular projection neurons to the CA-LP and PTTH neurons. This might be relevant for larvae where ecdysone or juvenile hormone would be secreted in response to olfactory cues, although the function of such a pathway is not known. We also reveal parallel paths from sensory to the mushroom body and lateral horn, through additional layers of interneurons (which include mushroom body output neurons), and onto the CA-LP and PTTH neuroendocrine targets (*Figure 6—figure supplement 3*).

Finally, we illustrate the key features of the neuronal circuit architecture that underlie the neuroendocrine system, which can be constructed using CRZ as an exemplary RPN (single output cell) (*Figure 7*). We start with the strongest connection from interneuron Munin 2 (#10), which receives input from a group of pharyngeal sensory neurons (*Figure 7*, panel 1). A second interneuron SiB (#12) receives input from a group of enteric sensory neurons (*Figure 7*, panel 2); this interneuron also receives inputs from a different class of pharyngeal sensory neurons. More interneurons are added to build a series of parallel paths (diverging sensory signals) that all converge on a common RPN (*Figure 7*, panel 3). These interneurons concurrently target different RPNs (*Figure 7*, panel 4; see also figure legend for details). At this point, then, a set of distinct RPNs becomes inexorably linked as the interneurons that converge onto the single CRZ neuron are also monosynaptically connected to other RPNs. Thus, the parallel paths that converge on a single RPN (e.g., CRZ) additionally target multiple RPNs, thereby forming a set of linked outputs. For single-cell networks of all RPNs, see *Figure 7—figure supplement 1*.

## Discussion
### The neuroendocrine connectome of *Drosophila* larvae

Organisms differ in their adaptive capacity to deal with external and internal changes, but the essential goal remains the same: ensuring homeostasis in a changing environment. Evolution of neuroendocrine systems led to the separation of sensory systems, neuroendocrine cells, and specialized glands (*Hartenstein, 2006*). We show in this paper how the central neuroendocrine system is synaptically organized. A general feature of the ring gland projection neurons (RPNs) is the absence of synaptic outputs within the CNS. The exception are the EH-producing neurons, which have synaptic outputs in the protocerebrum,

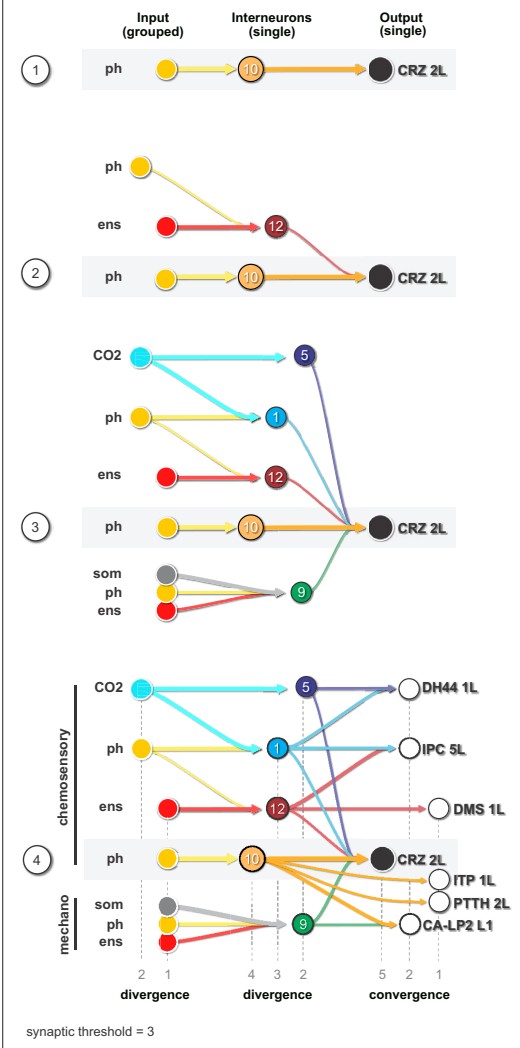

**Figure 7.** Diverging sensory and interneuronal paths converge onto a linked set of output neurons. Circuit architecture common for all RPNs (CRZ single-cell example). **1**: the strongest polysynaptic path based on hub analysis from pharyngeal sensory origin to CRZ output neuron via interneuron 10. **2**: second interneuron (12) integrating enteric information and different pharyngeal information, converging onto CRZ output neuron. **3**: all interneurons of one CRZ output neuron integrating multiple sensory origins and converging onto one single output. **4**: concept of divergence and convergence in the neuroendocrine connectome. Sensory neurons diverge/converge onto distinct sets of interneurons. Interneurons diverge in varying synaptic strength onto distinct sets of linked RPN output neurons. Numbers at bottom show degree of convergence and divergence (e.g., interneuron 10 diverges to CRZ, ITP, PTTH, and CA-LP; all interneurons converge to CRZ; synaptic threshold = 3 for all connections). CA-LP: corpus allatum innervating neurosecretory neurons of the lateral protocerebrum; $CO_2$: carbon dioxide; CRZ: corazonin neurons; DH44: diuretic hormone 44 neurons; DMS: *Drosophila*

*Figure 7 continued on next page*

*Figure 7 continued*

myosuppressin neurons; ens: enteric nervous system; IPCs: insulin-producing cells; ITP: ion transport peptide neurons; ph: pharynx; PTTH: prothoracicotropic hormone neurons; som: somatic; RPNs: ring gland projection neurons.

The online version of this article includes the following figure supplement(s) for figure 7:

**Figure supplement 1.** Diverging single-cell RPN circuits.

SEZ, and VNC. This unique feature of EH neurons might be due to their function in coordinating movements during larval cuticle shedding (*Baker et al., 1999*; *Krüger et al., 2015*). Another feature is that the RPNs of the PL are connected with the sensory organs exclusively via polysynaptic paths, which is in contrast to the numerous monosynaptic connections found for RPNs of the PI (*Miroschnikow et al., 2018*; *Schlegel et al., 2016*). It is also noteworthy that peptides known for their roles in metabolic and stress regulation in general receive large amounts of their inputs from interneurons with direct contacts to the sensory system, that is, these paths are short, with only a single hop between the interneurons and sensory neurons. This might be due to the need for rapid action compared to those (e.g., PTTH and CA-LP neurons) involved in gradual, long-term and irreversible events such as larval growth and maturation.

## Novel $CO_2$-responsive sensory to endocrine pathways: from connectomic-based modeling to in vivo testing

Numerous previously unknown synaptic pathways from the sensory organs to the RPNs were revealed from our connectomic analysis, including a new set of sensory neurons, namely the TD ($CO_2$) neurons that respond to $CO_2$ levels. This might be due to the stress associated with high levels of $CO_2$, which is observed in humans as well (*Permentier et al., 2017*). These sensory neurons target, via thoracic interneurons, RPNs that express two peptides known to play a dominant role in metabolic stress regulation in *Drosophila*: Dh44 and Crz (*Cannell et al., 2016*; *Dus et al., 2015*; *Kubrak et al., 2016*). From a neuronal network perspective, it was possible to predict this modulation with a feed forward network (FFN). Both peptide groups display homology to mammalian neuroendocrine axes known to regulate stress (HPA axis) and reproductive behavior (HPG axis). Dh44 is a homolog of vertebrate CRH, which is released in

the hypothalamus in response to external and internal stressors like hypoxia or hypoglycemia (*Flanagan et al., 2003*). A role for Dh44 in glucose and amino acid sensing has been reported (*Dus et al., 2015*; *Yang et al., 2018*), but its role in $CO_2$-dependent response was not previously known. $CO_2$ activation of Crz, a homolog to GnRH, adds to the repertoire of stress sensors ascribed to these neurons that include their roles in glucose and fructose sensing (*Dus et al., 2015*; *Kubrak et al., 2016*; *Miyamoto and Amrein, 2014*; *Oh et al., 2019*; *Veenstra, 2009*). Interestingly, this sensory pathway, which originates in the trachea, appears to be largely distinct from the $CO_2$ sensing pathway that originates in the terminal organ (*Faucher et al., 2006*). The physiological and behavioral consequences of $CO_2$-dependent response in the trachea, for example, whether it is appetitive or aversive, remain to be investigated. The connectome analysis further indicates that CRZ and DH44 neurons have the strongest synaptic connections with the sensory system (i.e., greatest number of paths that are connected monosynaptically or via single interneurons), suggesting a critical role of these neurosecretory cells in rapid sensory integration.

## Combinatorial parallel pathways enable variability and flexibility in the central neuroendocrine system

Sensory pathways are often studied based on a single type of sensory organ or modality, in most cases for technical reasons. In a natural environment, it is unlikely that an animal will encounter a situation where it needs to react to only a single sensory input and secrete a single type of hormone. For the fly larvae, two broad types of actions have to be taken into account: immediate action to an acute stress (e.g., due to toxic smoke, predator wasp, or starvation), and a slower action that enables tissue and organismal growth (e.g., accumulation of biosynthetic resources for cell growth and progression onto the next moulting or puparium stage). Even an acute response takes place within the existing physiological state of the organism. For the endocrine organs, this requires the secretion of different combinations, and most likely different concentrations, of hormones and neuropeptides into the circulation or target tissues.

At the core of the neuroendocrine network is a parallel set of interneurons that target distinct combinations of neuroendocrine outputs (e.g., the RPNs, each expressing certain neuropeptides). Each of the interneurons in turn receive sensory inputs from distinct sets of sensory neurons (e.g., $CO_2$ sensitive in trachea or different type and modality within the pharyngeal region). This can be also seen in the pathways from olfactory sensory neurons to CA-LP and PTTH endocrine targets. Multiglomerular projection neurons integrate olfactory as well as gustatory information, and as one proceeds deeper into the neuronal circuitry, interneurons that have originally been classified as interneurons without sensory input can be connected by additional hops to sensory neurons (such as through mushroom body and lateral horn in the protocerebrum). These then converge together with the multiglomerular projection neurons onto the common set of interneurons that target the CA-LP and PTTH output neurons. The different converging paths can be seen to represent distinct types of sensory information, including a stored form from the mushroom body (*Eichler et al., 2017*; *Eschbach and Zlatic, 2020*; *Miroschnikow et al., 2020*), where a positive or negative valence has been attached to an existing sensory cue. Additionally, there are a significant number of synaptic connections among the interneurons. Such architecture would enable variability and flexibility in the combination and concentrations of neuropeptides that become released in response to the flood of multisensory inputs that act on all parts of the neuroendocrine network. Subsequently cross-regulatory interactions at the receptor level would then determine the final neuropeptide/hormone composition that is released within the CNS or into the circulation. Our work provides a neuronal architectural blueprint of how this is constructed at the synaptic level for the neuroendocrine system in the brain and may also be of general relevance in understanding other complex neuroendocrine systems.

As a concluding remark, the neuroendocrine connectome of the *Drosophila* larva presented here (i.e., the 'ring gland connectome') represents the first complete synaptic map of sensory to endocrine pathways in a neuroendocrine system of this complexity and adds another level of insight on the known humoral functions of the released neuropeptides and hormones. Together with the large amount of knowledge on the function of neurosecretory cells targeting the CC, CA, PG, and AO over the past years (*Nässel and Zandawala, 2020*), the current analysis increases our understanding of how the neuroendocrine system receives information about external and internal sensory cues. A future challenge in this context is the identification of specific sensory neurons of different origin and modality to define the valence of sensory integration, and the function of the interneurons that enable different pathways to the endocrine organs.

# Materials and methods

## Flies

All larvae used for experiments and stainings were 96 ± 4 hr (after egg laying) of age and were grown on standard cornmeal medium under a 12 hr light/dark cycle if not otherwise stated. The following driver and effector lines were used (also see *Table 1* for genotypes of experimental flies):

**Table 1.** Genotypes of experimental flies.

| Figure | Genotypes | Chr. |
|---|---|---|
| | *w; P{UAS-mCD8.mRFP.LG}18a; P{Ms-GAL4.P}* | X; 2; 3 |
| | *w; P{Ilp2-GAL4.R}2/ P{UAS-mCD8.mRFP.LG}18a* | X; 2 |
| | *w; P{UAS-mCD8.mRFP.LG}18a; P{Dh44-GAL4.TH}2 M* | X; 2; 3 |
| | *w; P{UAS-mCD8.mRFP.LG}18a; P{Crz-GAL4.391}4 M* | X; 2; 3 |
| | *w; TI{2A-lexA::QF}ITP[2A-lexA.no1]; P{13XLexAop2-IVS-myr::GFP}attP2* | X; 2; 3 |
| | *w; P{Burs-GAL4.TH}4 M/ P{UAS-mCD8.mRFP.LG}18a* | X; 2 |
| | *w; P{UAS-mCD8.mRFP.LG}18a; P{Ptth-GAL4.M}45, P{Ptth-GAL4.M}117b* | X; 2; 3 |
| | *w; P{UAS-mCD8.mRFP.LG}18a; P{GMR17B03-GAL4}attP2* | X; 2; 3 |
| | *w; PBac{IT.GAL4}CG7997[0714-G4]/ P{UAS-mCD8.mRFP.LG}18a* | X; 2 |
| *Figure 3C* (antibody staining, from left to right) | *w; P{GAL4-Eh.2.4}C21P/ {UAS-mCD8.mRFP.LG}18a* | X; 2 |
| | *w; P{Ms-GAL4.P}/ PBac{UAS-CaMPARI2}VK00005* | X; 3 |
| | *w; P{Ilp2-GAL4.R}2; PBac{UAS-CaMPARI2}VK00005* | X; 2; 3 |
| | *w; P{Dh44-GAL4.TH}2 M/ PBac{UAS-CaMPARI2}VK00005* | X; 3 |
| | *w; P{Crz-GAL4.391}4 M/ PBac{UAS-CaMPARI2}VK00005* | X; 3 |
| | *w; TI{2A-GAL4}ITP[2A-D.GAL4]; PBac{UAS-CaMPARI2}VK00005* | X; 2; 3 |
| | *w; P{Ptth-GAL4.M}45, P{Ptth-GAL4.M}117b/ PBac{UAS-CaMPARI2}VK00005* | X; 3 |
| | *w; P{GMR17B03-GAL4}attP2/ PBac{UAS-CaMPARI2}VK00005* | X; 3 |
| | *w; PBac{IT.GAL4}CG7997[0714-G4]; PBac{UAS-CaMPARI2}VK00005* | X; 2; 3 |
| *Figure 4B* (CaMPARI analysis, from top left to bottom right) | *w; P{GAL4-Eh.2.4}C21P; PBac{UAS-CaMPARI2}VK00005* | X; 2; 3 |
| *Figure 2—figure supplement 1B,C* | Same genotypes as in *Figure 3C* | |
| *Figure 2—figure supplement 2B* | *w; TI{2A-lexA::QF}ITP[2A-lexA.no1]; P{13XLexAop2-IVS-myr::GFP}attP2* | X; 2; 3 |
| *Figure 2—figure supplement 3B* | *w; P{UAS-mCD8.mRFP.LG}18a; P{Crz-GAL4.391}4 M* | X; 2; 3 |
| *Figure 2—figure supplement 3C* | *w; P{CrzR-GAL4.3.5.S}T11A/ P{UAS-mCD8.mRFP.LG}18a* | X; 2 |
| *Figure 2—figure supplement 3D* | *y, w, P{UAS-myrGFP.QUAS-mtdTomato-3xHA}su(Hw)attP8; P{trans-Tango}attP40; P{Crz-GAL4.391}4 M* | X; 2; 3 |
| *Figure 3—figure supplement 2A* | *w; P{UAS-mCD8.mRFP.LG}18a; PBac{IT.GAL4}lqfR[0260-G4]* | X; 2; 3 |
| *Figure 3—figure supplement 2B,C* | *w; P{UAS-CaMPARI}attP40; PBac{IT.GAL4}lqfR[0260-G4]* | X; 2; 3 |
| *Figure 3—figure supplement 2D* (antibody staining, from left to right) | *w; P{Gr21a-Mmus\Cd8a.GFP}2* | X; 2 |
| | *w; P{Gr21a-GAL4.C}133t52.1/ P{10XUAS-mCD8::GFP}attP2* | X; 3 |
| | *w; P{Gr21a-GAL4.C}133t1.2/ P{10XUAS-mCD8::GFP}attP2* | X; 3 |

*Ilp2-Gal4* (IPC neurons, BL#37516), *Ms-Gal4* (DMS neurons, **Park et al., 2008**), *Dh44-Gal4* (DH44 neurons, BL#51987), *Crz-Gal4* (CRZ neurons, BL#51977), *CrzR-Gal4$^{T11A}$* (**Sha et al., 2014**), *Ptth-Gal4* (PTTH neurons, **McBrayer et al., 2007**), *Burs-Gal4* (BL#51980), *Burs-Gal4* (BL#40972, this line shows expression in CA-LP neurons of the PL, data not shown), *Eh-Gal4* (EH neurons, BL#6301), *17B03-Gal4* (Hug$^{RG}$ neurons, **Jenett et al., 2012**), *714*-Gal4 (CAPA neurons, **Gohl et al., 2011**), *ITP-T2A:Gal4* (ITP neurons, used in CaMPARI experiments, unspecific expression in CNS glia observed BL#84702), *ITP-T2A:LexA* (ITP neurons, for generation see below, used in stainings – clean expression of ITP in the CNS), *260*-Gal4 (TD $CO_2$ neurons, BL#62743), *Gr21a-Gal4* (BL#23890, BL#24147), *Gr21a-GFP* (BL#52619), *UAS-mRFP* (BL#27398), *UAS-CaMPARI-1* (BL#58761), *UAS-CaMPARI-2* (BL#78316), *UAS-GFP* (BL#32184), *trans-Tango* (BL#77124), and *lexAop2-myrGFP* (BL#32209).

## Generation of *ITP-T2A-LexA* transgenic fly lines

First, we generated *T2A-LexA:QF* knock-in constructs that can be targeted to genomic loci by homology-directed repair using the CRISPR/Cas system. Therefore, *splice acceptor-T2A-LexA:QF* fragments for all three intron phases were amplified by PCR (Q5 polymerase, New England Biolabs) from *pBS-KS-attB2-SA(0/1/2)-T2A-LexA::QFAD-Hsp70* plasmids (Addgene #62947, #62,948, and #62949) (**Diao et al., 2015**) with primers CGTACTCCACCTCACCCATC and ctcgagAAGCTTCT GAATAAGCCCTCGT. PCR products were sub-cloned into *pCRII-TOPO* vector (Invitrogen) to create plasmids *TOPO-T2A-LexA:QF(0)*, *TOPO-T2A-LexA:QF(1)*, and *TOPO-T2A-LexA:QF(2)*. Next *splice acceptor-T2A-Gal4* cassette from *pT GEM(0)* (Addgene #62891) (**Diao et al., 2015**) was removed by *XbaI/SalI* digest and replaced with *XbaI/XhoI* fragments from *TOPO-T2A-LexA:QF(0)*, *TOPO-T2A-LexA:QF(1)*, and *TOPO-T2A-LexA:QF(2)* harboring *splice acceptor T2A-LexA:QF* cassettes (T-LEM, T2A-LexA expression module) for all three intron phases. All restriction enzymes used and T4 DNA ligase are from New England Biolabs. We named these *T2A-LexA:QF* knock-in plasmids *pT-LEM(0)*, *pT-LEM(1),* and *pT LEM(2)*.

Two CRISPR target sites (no1 and no2) in the intron downstream of the first coding exon shared by all five predicted transcripts of the *Ion transport peptide* gene (*ITP*) to insert T-LEM were chosen using *flyCRISPR Optimal Target Finder* (**Gratz et al., 2014**). By ligating annealed oligonucleotides, two guide RNA expression constructs were inserted into *BbsI*-linearized *pCFD3* vector (**Port et al., 2014**). Sequences of oligonucleotides were.

> (no1) gtcgGTGTTCCTTACAGCGTTCA aaacTGAACGCTGTAAGGAACAC.
> (no2) gtcgAAAATGATCGCGGGACCTT aaacAAGGTCCCGCGATCATTTT.

Next, 5prime and 3prime homology arms (5´HA, 3´HA) for both targeted sites were introduced into *pT-LEM(2)*. Therefore, target site flanking sequences of approximately 1 kb size were amplified by PCR (Q5 polymerase, New England Biolabs) from genomic DNA of *nos-Cas9$^{[attP2]}$* fly line used for embryo injection. See *Table 2* for primer sequences. PCR products were subcloned into *pCRII-TOPO* vector (Invitrogen). Then 5´HAs were ligated as *SphI/NotI* fragments from TOPO plasmids into *SphI/NotI*-linearized *pT-LEM(2)* vector, resulting in *pT-LEM(2)–5´HA-no1* and *pT-LEM(2)–5´HA-no2*. Finally, 3´HA no1 was inserted as *AscI/KpnI* fragment from TOPO plasmid into *AscI/KpnI*-digested *pT-LEM(2)– 5´HA-no1* and 3´HA no2 as *KpnI/SpeI* fragment into *KpnI/SpeI*-cut *pT-LEM(2)–5´HA-no2*, resulting in *pT-LEM(2)-ITP-no1* and *pT-LEM(2)-ITP-no2*, respectively. Plasmid microinjections to generate *ITP$^{T2A-LexA-no1}$* and *ITP$^{T2A-LexA-no2}$* lines were performed by BestGene Inc By using Cre-loxP system, the 3xP3-DsRed cassette was removed from *ITP$^{T2A-LexA-no1}$* and *ITP$^{T2A-LexA-no2}$*.

**Table 2.** Primer sequences to generate homology arms.

|  | Forward primer sequence | Revers primer sequence |
| --- | --- | --- |
| 5´HA no1 | gcatgcACGCGCTGTTAATCAAAT | gcggccgcACGCTGTAAGGAACACTGATG |
| 5´HA no2 | gcatgcCGCTGTCATCGCTGTAATTC | gcggccgcGTCCCGCGATCATTTTCC |
| 3´HA no1 | ggcgcgccTCAAGGCAAGGCCGTCC | ggtaccCGAATTAAATTTGGGCGTTT |
| 3´HA no2 | ggtaccCTTCGGTTGTTTCTGAACTTTATG | actagtTCTCCCACTCCCCAATTATG |

## EM reconstruction

Neuron reconstruction was done on an ssTEM volume of a 6-hr-old first instar larva (*Ohyama et al., 2015*). We identified the RPNs by reconstruction of all axons originating in the CNS and targeting the ring gland through the NCC nerve. The mNSCs including neurons producing insulin-like peptides, DMS and DH44, have been previously reconstructed and described (*Miroschnikow et al., 2018*; *Ohyama et al., 2015*; *Schlegel et al., 2016*). We reconstructed all neurons to completion (tracing 100% and at least 95% reviewed). Downstream targets were not synaptically connected to RPNs (except for EH downstream partners, being reconstructed with a synaptic threshold of 3). Therefore, membrane-fused DCVs were marked as connectors without direction. DCVs within the CNS were not marked due to technical issues with the common synapse annotation system. No synaptic connections were observed in the larval ring gland. The ring gland was reconstructed with all cells and tissue areas were assigned based on tissue boundaries, color (CA area was slightly darker, CC cells showed dendritic arborizations into the CC), and cell soma position. All synaptic up- and downstream partners of the RPNs were reconstructed to completion with a synaptic threshold to each of the RPNs of three synapses.

For sensory neurons included here, we made use of earlier published data (*Berck et al., 2016*; *Miroschnikow et al., 2018*; *Ohyama et al., 2015*; *Schlegel et al., 2016*). A subset of 12 TD neurons were already described (*Schlegel et al., 2016*). We reconstructed for this study all 26 TD neurons.

## Sensory neuron pie charts

Pie charts in *Figure 3* and following: Pie charts of sensory profiles were calculated using the percentage of total synaptic input of interneurons and RPNs (in case of monosynaptic connections) as fraction (thereby ignoring other inputs to show distribution of sensory origins). Percentages then give the percentage of total sensory synaptic input to interneurons or RPNS.

## Hub score

Calculation of hub score in *Figure 5A*: Fraction of total synaptic input from all sensory neurons to defined interneurons (see IDs) was multiplied by the total fraction of input of the RPN group from this interneuron. For example, interneuron #10 (Munin 2) receives 32.33% (fraction: 0.3233) of their total synaptic input from sensory neurons. In turn, corazonin neurons receive 56.52% (fraction: 0.5652) of their total synaptic inputs from interneuron #10 (Munin 2). Multiplying the fractions of this path (sensory via interneuron to CRZ) leads to a hub score of $0.3233 \times 0.5652 = 0.18272916$ (hub score).

## Immunohistochemistry

Dissected larval brains were fixed for 1 hr in paraformaldehyde (4%) in 1× phosphate-buffered saline (PBS), rinsed three times (20 min) with 1% PBS-T (1% Triton X-100 in 1× PBS), and blocked in 1% PBS-T containing 5% normal goat serum (ThermoFisher) for 1 hr. Primary antibody was added to the solution (for concentrations, see below). Brains rotated overnight at 4°C. On the second day, larval brains were washed three times (20 min) with 1% PBS-T and subsequently secondary antibody was applied. Brains rotated overnight at 4°C. After three times washing with 1% PBS-T, brains were dehydrated through an ethanol-xylene series and mounted in DPX Mountant (Sigma-Aldrich). Imaging was carried out using a Zeiss LSM 780 confocal microscope with 25× or 63× objective (oil). For antibody stainings of *peptide> mRFP*, the primary antibody was anti-RFP (1:500, mouse, Abcam, ab65856). Secondary antibody was anti-Mouse Alexa Fluor 568 (1:500, goat, Invitrogen, A-11031). For *ITP>myr* GFP stainings, primary antibody was anti-GFP (1:500, chicken, Abcam, ab13970) and secondary antibody was anti-Chicken Alexa Fluor 488 (1:500, goat, Invitrogen, A-11039). For Crz staining, primary antibody was anti-Crz (1:500, rabbit, gift from C. Wegener), secondary antibody was anti-Rabbit Alexa Fluor 568 (1:500, goat, Invitrogen, A-11011). For Trans-Tango stainings, primary antibodies were anti-GFP (1:500, chicken, Abcam, ab13970) and anti-HA (1:250, mouse, BioLegend, 901501). Secondary antibodies were anti-Chicken Alexa Fluor 488 (1:500, goat, Invitrogen, A-11039) and anti-Mouse Alexa Fluor 568 (1:500, goat, Invitrogen, A-11031), respectively. For Gr21a>GFP stainings, primary antibodies were anti-GFP (1:500, chicken, Abcam, ab13970) and anti-Futsch/22C10 (1:500, mouse, DSHB, AB528403). 22C10 was deposited to the DSHB by S. Benzer and N. Colley. Secondary antibodies were anti-Chicken Alexa Fluor 488 (1:500, goat, Invitrogen, A-11039) and anti-Mouse Alexa Fluor 633 (1:500, goat, Invitrogen, A-21046), respectively. DAPI (1:1,000) was used for staining of RG nuclei.

## Functional imaging with CaMPARI

For experiments with TD-neuron line *260*-Gal4, we used *UAS-CaMPARI1* (*Fosque et al., 2015*). A larva was placed inside the Petri dish and fixed with duct tape for 60 s. 405 nm UV light (M405L2_UV, Thorlabs) was placed 12 cm above the larva and illuminated with a LED controller (LEDD1B, Thorlabs at max intensity) for 15 s. Afterwards the larval brain was dissected and put onto a poly-L-lysine-coated coverslip and covered with 1× PBS for imaging at low $Ca^{2+}$ conditions. Caudal dendrites of TD neurons that project to the SEZ were imaged. For defined concentrations of $CO_2$ stimulation, we used a $CO_2$ incubator (CB 53, Binder) at $CO_2$ concentrations of 0, 10, and 20% $CO_2$ at 24–27°C. Stimulation protocol was the same as described before.

For experiments with different *peptide-Gal4* lines, we used *UAS-CaMPARI2* with improved baseline fluorescence and improved integration dynamics (*Moeyaert et al., 2018*). In our hands, photoconversion ratios were lower in general but more defined when neurons were not active, lowering the number of false-positive photoconversion (own observations). We used the $CO_2$ incubator to set $CO_2$ concentration to 20% and compared neuronal photoconversion with 0% $CO_2$ concentration in the incubator. Larvae were placed on duct tape in the middle of a 5 cm Petri dish for 60 s and afterwards illuminated for 30 s with 405 nm at max intensity. Following steps were the same as described before.

## Statistics

For CaMPARI experiments, green to red ratios of single cells of *peptide-Gal4* lines were analyzed with a custom-made script for FIJI (ImageJ), and the mean was calculated per animal (each cell was analyzed and a mean build). Animal means were then analyzed and plotted with Prism 6 software using the Mann–Whitney rank-sum test, $*p<0.05$, $**p<0.01$, $***p<0.001$, $****p<0.0001$.

## FFN diffusion model

The FFN was implemented in Python as a simple artificial neural network without backpropagation. Synaptic weights were normalized by the total number of postsynapses such that they represented fractions of inputs for a given neuron. Neurons were implemented as rectified linear units using a ReLu activation function that starts responding at 5% and reaches saturation at 50% of their synaptic inputs being active:

$$\mathrm{f}(x) = \begin{cases} 0 & if\, x < a \\ (x - a)/(b - a) & if\, a < x < b \\ 1 & if\, x > b \end{cases}$$

with x being the sum activity of all inputs weighted by their synaptic weights, constants *a* and *b* controlling the response onset and saturation, respectively. *a* and *b* were chosen such that neurons start responding at 5% and reach saturation at 50% of their synaptic inputs being active: *a* = 0.05, *b* = 0.5. These values were chosen to maximize the response range of the network. The code for the FFN and the generation of the figures can be found at https://github.com/Pankratz-Lab/FFN_Hueckesfeld-et-al.-2020 (*Schoofs and Schlegel, 2020*; copy archived at swh:1:rev:494220124eb79f5ed0b5eebe585b796e18729b47).

## Analysis of single-cell transcriptomic data from *Brunet Avalos et al., 2019*

In order to analyze peptide receptor interaction between RPN groups, we sought out to use the data generated in the lab of Simon Sprecher describing the single-cell transcriptomic atlas of the *Drosophila* larval brain (*Brunet Avalos et al., 2019*). Advantage of this dataset was the exclusive analysis of SEZ and brain lobes, which helped in finding the RPN-specific peptidergic cell groups. We used R analysis similar to the described workflow in *Brunet Avalos et al., 2019* based on Seurat v3 workflow (*Butler et al., 2018*; *Stuart et al., 2019*). In brief, we used seurat processing pipeline from Satija lab (https://satijalab.org/seurat/) to process the integrated datasets of fed and starved conditions (GEO accession number GSE134722 *Brunet Avalos et al., 2019*). This combined dataset consists of 9346 cells and 14,064 analyzed features. In order to cluster the RPNs into the specific groups, the following parameters were used: dataset: fed and starved integrated and log normalized | scale = 10,000 | 2000 variable genes | Seurat v3 processing: cells with unique features: 200–4500 |

genes expressed in at least one cell | 31 PCs were used to assess cell clusters | resolution was 1 | cluster 12 was identified as peptidergic cells | peptidergic cells were separated with the following parameters (expression profiles):

IPCs: Ilp2 ≥ 3 & Ilp5 & Ilp3 (26 cells)
DMS: Ms ≥ 6 (9 cells)
DH44: Dh44 ≥ 2.8 (12 cells)
CRZ: Crz ≥ 1 &sNPF ≥ 1 (13 cells)
ITP: ITP ≥ 1 & Lk ≥ 0.8 (17 cells)
PTTH: Ptth ≥ 2 (9 cells)
CA-LP: FMRFa >3.5 (14 cells)
Hug$^{RG}$: Hug >4 & Mip > 1 (7 cells)
CAPA: Capa ≥ 2 (6 cells)
EH: Eh ≥ 4 (4 cells)

For CA-LP neurons, FMRFamide was used based on the description in *de Velasco et al., 2007*. Hugin-RG cells were separated based on Coexpression of Mip neuropeptide (unpublished observation, staining with Mip-Gal4 line and Hugin-antibody).

## Graphical representation and visualization

Neurons were rendered in Blender 3D (ver2.79b) using the CATMAID to Blender interface described by *Schlegel et al., 2016* (https://github.com/elifesciences-publications/Catmaid-to-Blender) and edited in Affinity Designer (Serif) for MAC. Staining images were processed with FIJI (ImageJ) and CaMPARI images were analyzed using a custom-made FIJI script to be subsequently edited in Affinity Designer. Hive Plots were generated by using the CATMAID software for spatial distribution of nodes and subsequently made in Gephi 0.92 with rescaled edge weights (e.g., 1–200 synapses were rescaled for line thickness 1–20). Edges with less than five synapses were ignored in Gephi. To visualize peptide receptor connectivity, we used Circos tableviewer (http://mkweb.bcgsc.ca/tableviewer/).

## Acknowledgements

We thank the whole Pankratz lab for fruitful discussions on earlier versions of this manuscript. We thank Anna Pepanian, Christina Georgopoulou, and Stephan Nottelmann for help with cloning of the ITP-T2A-LexA line. We thank Dick Nässel, Michael B O´Connor, Jae Park, Yi Rao, Jan Veenstra, Christian Wegener, and Benjamin White for sharing fly lines and resources. We thank the Fly EM Project Team at HHMI Janelia for the gift of the EM volume, the HHMI visa office, and HHMI Janelia for funding. We also thank all 'tracers' for their contribution to the EM reconstruction.

## Additional information

### Funding

| Funder | Grant reference number | Author |
| --- | --- | --- |
| Deutsche Forschungsgemeinschaft | DFG PA 787/9-1 | Sebastian Hückesfeld<br>Philipp Schlegel<br>Anton Miroschnikow<br>Andreas Schoofs<br>Ingo Zinke<br>André N Haubrich<br>Michael J Pankratz |
| Howard Hughes Medical Institute | | Casey M Schneider-Mizell<br>James W Truman<br>Richard D Fetter<br>Albert Cardona |
| Wellcome Trust | 205038/Z/16/Z | Albert Cardona |

| Funder | Grant reference number | Author |
|--------|------------------------|--------|
| Deutsche Forschungsgemeinschaft | EXC 2151 - 390873048 | Sebastian Hückesfeld<br>Philipp Schlegel<br>Anton Miroschnikow<br>Andreas Schoofs<br>Ingo Zinke<br>André N Haubrich<br>Michael J Pankratz |

The funders had no role in study design, data collection and interpretation, or the decision to submit the work for publication.

## Author contributions

Sebastian Hückesfeld, Conceptualization, Data curation, Formal analysis, Investigation, Methodology, Supervision, Validation, Visualization, Writing – original draft, Writing – review and editing; Philipp Schlegel, Conceptualization, Data curation, Formal analysis, Investigation, Methodology, Software, Writing – original draft, Writing – review and editing; Anton Miroschnikow, Conceptualization, Data curation, Formal analysis, Investigation, Methodology, Validation, Visualization, Writing – review and editing; Andreas Schoofs, Investigation, Validation, Writing – review and editing; Ingo Zinke, Data curation, Investigation, Methodology, Validation, Visualization, Writing – review and editing; André N Haubrich, Investigation, Methodology, Writing – review and editing; Casey M Schneider-Mizell, Richard D Fetter, Methodology, Resources, Software, Writing – review and editing; James W Truman, Investigation, Resources, Writing – review and editing; Albert Cardona, Investigation, Methodology, Resources, Software, Supervision, Writing – review and editing; Michael J Pankratz, Conceptualization, Formal analysis, funding-acquisition, Investigation, project-administration, Resources, Supervision, Visualization, Writing – original draft, Writing – review and editing

## Author ORCIDs

Sebastian Hückesfeld ⓘD http://orcid.org/0000-0003-0236-6375
Philipp Schlegel ⓘD http://orcid.org/0000-0002-5633-1314
Anton Miroschnikow ⓘD http://orcid.org/0000-0002-2276-3434
Andreas Schoofs ⓘD http://orcid.org/0000-0001-7002-9181
André N Haubrich ⓘD http://orcid.org/0000-0001-7895-6203
Casey M Schneider-Mizell ⓘD http://orcid.org/0000-0001-9477-3853
James W Truman ⓘD http://orcid.org/0000-0002-9209-5435
Richard D Fetter ⓘD http://orcid.org/0000-0002-1558-100X
Albert Cardona ⓘD http://orcid.org/0000-0003-4941-6536
Michael J Pankratz ⓘD http://orcid.org/0000-0001-5458-6471

## Decision letter and Author response

Decision letter https://doi.org/10.7554/eLife.65745.sa1
Author response https://doi.org/10.7554/eLife.65745.sa2

# Additional files

### Supplementary files

• Supplementary file 1. Statistics table. Green to red ratios in CaMPARI measurements. For experiments and use of statistical tests, see Materials and methods.

• Transparent reporting form

### Data availability

All data generated or analysed during this study are included in the manuscript and supporting files. We used the same EM volume reported in Ohyama et al. 2015 (Nature) and available at https://neurodata.io/data/acardona_0111_8. To access the dataset, users need to first create a free account on the neurodata site: the data is then subsequently available to download (further details can be found in the guide https://neurodata.io/help/download/). There are no restrictions on availability. The following previously published data sets were used: Ohyama T Schneider-Mizell CM Fetter RD Valdes Aleman J Franconville R Rivera-Alba M Mensh BD Branson KM Simpson JH Truman JW (2015)

NeuroData EM volume from: A multilevel multimodal circuit enhances action selection in *Drosophila*. https://neurodata.io/data/acardona_0111_8.

The following previously published datasets were used:

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
