## [Decision Letter]

**Acceptance summary:**

In this manuscript, the authors use a serial section transmission electron microscopy data of a *Drosophila* larva to reconstruct its complete neuroendocrine system – from sensory inputs to neuroendocrine cell outputs. It highlights the complexity of brain circuits, describing how parallel processing systems can lead to a multitude of different input combinations for different neuroendocrine cell types and subcircuits.

**Decision letter after peer review:**

Thank you for submitting your article "Unveiling the sensory and interneuronal pathways of the neuroendocrine connectome in *Drosophila*" for consideration by *eLife*. Your article has been reviewed by 3 peer reviewers, including Sonia Sen as the Reviewing Editor and Reviewer #1, and the evaluation has been overseen by K VijayRaghavan as the Senior Editor. The following individual involved in review of your submission has agreed to reveal their identity: Kim Rewitz (Reviewer #3).

Essential revisions:

1. Our major concern is whether or not the new CO2 circuit the authors describe overlaps with the known GR21a CO2 circuit. To sort this out, our recommendations are:

– Identify and map the GR21a circuit in the EM data and show if/where it intersects with the tracheal CO2 circuit.

– Show more cleanly whether it's only the tracheal neuron that are activating the neuropeptide neurons (and not the GR21a neurons too). They could do this by activating the TD neurons artificially in their CAMPARI set up if they have clean genetic access to these neurons. If they don't, this experiment need not be done – the GR21a circuit tracing in EM will be sufficient.

– A clean line *does* exist for Gr21a. So, the authors could test whether or not this CO2 sensory circuit activates the neuropeptide neurons in the CAMPARI set-up by inactivating these neurons while monitoring the neuropeptide neurons.

2. Analyze the EM data with a stable synaptic threshold for connectivity, or explain why there are different thresholds for sensory and interneuron connectivity (2 or 3). In the literature, other studies often use a threshold of 5. Also, the visualization graphs exclude all edges with less than 5 synapses, maybe the authors can use the same thresholds here as they do for analysis.

Recommendations for the authors:

– It will help the reader to have more descriptive text in the Results section. As the manuscript is currently written, much of the information is in figure legends. Often only a single sentence is used to summarise entire sub-figures that are rich in their findings and implications.

– Some results are tucked away in the supplementary information that I thought could be brought forward into the main manuscript, for example, the CaMPARI validation of the CO2 circuit and parts of the modelling and connectivity hub analysis.

– Maybe the authors can state more clearly if their connectivity analyses also includes connections between interneurons (do they express neuropeptide receptors? do they talk to each other?), and feedback connections back to the sensory system (Are there any?)?

---

## [Author Response]

Essential revisions:1. Our major concern is whether or not the new CO2 circuit the authors describe overlaps with the known GR21a CO2 circuit.To sort this out, our recommendations are:– Identify and map the GR21a circuit in the EM data and show if/where it intersects with the tracheal CO2 circuit.– Show more cleanly whether it's only the tracheal neuron that are activating the neuropeptide neurons (and not the GR21a neurons too). They could do this by activating the TD neurons artificially in their CAMPARI set up if they have clean genetic access to these neurons. If they don't, this experiment need not be done – the GR21a circuit tracing in EM will be sufficient.– A clean line does exist for Gr21a. So, the authors could test whether or not this CO2 sensory circuit activates the neuropeptide neurons in the CAMPARI set-up by inactivating these neurons while monitoring the neuropeptide neurons.

To address this point, we now provide several new analyses. First, we performed expression analysis with three different promoter-lines of Gr21a to see if these would show expression in the VNC indicative of tracheal dendritic (TD) neurons. All three were negative (in all cases expression in the SEZ was seen) (New Figure 3 – supplement 2, new panel D). Second, we performed connectivity modeling FFN analysis to see how the RPN outputs of known Gr21a neurons would compare with the TD CO_2_ responsive circuit. Faucher et al. 2006 (as referred to by Reviewer 2) showed that Gr21a drives expression in a single cell in the terminal organ. The major hurdle here is that although we can identify through which nerve bundle the Gr21a cells must project, we cannot unambiguously identify exactly which axonal projection belongs to the Gr21a expressing neuron. Thus, we used all neurons of that nerve bundle for the connectomic analysis to see what RPNs (ring gland projection neurons) are targeted, or more specifically, would DH44 and CRZ (the targets of tracheal CO_2_ circuit) also be reached. Our analysis indicated that only very weak connections are found to these RPNs, using different interneurons (New Figure 5—figure supplement 3). Taken together, these demonstrate that the two pathways are essentially independent, with little overlap.

2. Analyze the EM data with a stable synaptic threshold for connectivity, or explain why there are different thresholds for sensory and interneuron connectivity (2 or 3). In the literature, other studies often use a threshold of 5. Also, the visualization graphs exclude all edges with less than 5 synapses, maybe the authors can use the same thresholds here as they do for analysis.

To address this, we provide the following new analysis and explanations. We present a global view of the completeness of our work with respect to the different synaptic thresholds as applied to sensory and interneurons (New Figure 2—figure supplement 5). We end this new addition with the following paragraph:

“In subsequent analysis, synaptic thresholds were chosen based on the scientific message we want to convey in the clearest possible manner, and may differ depending on the type of graphs, schemes or tables used. […] One cannot say a priori which is more useful for a given figure, as the number of synapses and synaptic partners can vary widely depending on the neuron”.

For visualization of specific neurons, we chose different edge thresholds to emphasize the strongest connections. We also altered some of the visualization graphs in Figure 5 for clarity and consistency with other representations (New Figure 5, panel A).

Recommendations for the authors:– It will help the reader to have more descriptive text in the Results section. As the manuscript is currently written, much of the information is in figure legends. Often only a single sentence is used to summarise entire sub-figures that are rich in their findings and implications.

We have now put more details in the main text in addition to the legends (tracked changes).

As part of this effort to be less condensed, we have also separated the old figure into two figures (New and separated Figures 6 and 7).

– Some results are tucked away in the supplementary information that I thought could be brought forward into the main manuscript, for example, the CaMPARI validation of the CO2 circuit and parts of the modelling and connectivity hub analysis.

We have followed this advice and have moved the CaMPARI experiments into the main text (New figure 4). We have also moved a hub analysis from supplement to main figure (New Figure 6, panel A).

– Maybe the authors can state more clearly if their connectivity analyses also includes connections between interneurons (do they express neuropeptide receptors? do they talk to each other?), and feedback connections back to the sensory system (Are there any?)?

We have not emphasized the interactions between the interneurons, nor the feedback connections back to the sensory system. These certainly exist, but we thought including such analysis in this paper would somewhat distract from the main message: namely the short forward paths from the sensory to the interneurons and from the interneuron to the endocrine targets. In certain cases we show the interneuronal connections (New Figure 5A, Figure 6 – supplement 3; Figure 5 – supplement 2, panel B), and feedback connections to the sensory neurons (New Figure 6, panel A), but do not discuss these.

Finally, reviewer 2 made several points as part of “Weaknesses of the paper”, including investigating more on the different possible CO2 pathways, and discussing the fact that this is a single EM data set, and that these may be different across individuals and development. We have included more analysis and discussion on the Gr21a and the current pathway, as suggested, as this is indeed a novel point. For the other point, with all due respect, these points have been made in earlier papers on the larval volume: the weakness is indeed this is an n of 1, and first instar only, and it is known that projection changes occur during larval growth from first to third instar through light microscopy, as well as having differences across individual brains of same larval stage. Thus, we have chosen not to specifically mention these facts again.